# Novel Highly Efficient Antibacterial Chitosan-Based Films

**DOI:** 10.3390/biotech12030050

**Published:** 2023-07-07

**Authors:** Omar M. Khubiev, Anton R. Egorov, Nikolai N. Lobanov, Elena A. Fortalnova, Anatoly A. Kirichuk, Alexander G. Tskhovrebov, Andreii S. Kritchenkov

**Affiliations:** 1Faculty of Science, Peoples’ Friendship University of Russia (RUDN University), Miklukho-Maklaya St. 6, 117198 Moscow, Russia; ihubievomar1@gmail.com (O.M.K.); lobanov-nn@rudn.ru (N.N.L.); fortalnova-ea@rudn.ru (E.A.F.); kirichuk-aa@rudn.ru (A.A.K.); alexander.tskhovrebov@gmail.com (A.G.T.); 2Institute of Technical Acoustics NAS of Belarus, Ludnikova Prosp. 13, 210009 Vitebsk, Belarus

**Keywords:** chitosan, chitin, iron(III), click chemistry, antibacterial activity

## Abstract

In this study, we elaborated new chitosan-based films reinforced by iron(III)-containing chitosan nanoparticles Fe(III)-CS-NPs at different concentrations. We found that the optimum concentration of Fe(III)-CS-NPs for the improvement of antibacterial and mechanical properties of the films was 10% (σ_b_ = ca. 8.8 N/mm^2^, ε_b_ = ca. 41%, inhibition zone for *S. aureus* = ca. 16.8 mm and for *E. coli* = ca. 11.2 mm). Also, using the click-chemistry approach (thiol–ene reaction), we have synthesized a novel water-soluble cationic derivative of chitin. The addition of this derivative of chitin to the chitosan polymer matrix of the elaborated film significantly improved its mechanical (σ_b_ = ca. 11.6 N/mm^2^, ε_b_ = ca. 75%) and antimicrobial (inhibition zone for *S. aureus* = ca. 19.6 mm and for *E. coli* = ca. 14.2 mm) properties. The key mechanism of the antibacterial action of the obtained films is the disruption of the membranes of bacterial cells. The elaborated antibacterial films are of interest for potential biomedical and food applications.

## 1. Introduction

In the last few decades, there has been an increase in the number of studies devoted to the elaboration of antibacterial films for biomedical food applications [1,2,3]. Such films can be used in medicine for the treatment and healing of wounds, burns, caries, bones, and mucosal injuries [4,5,6,7,8,9,10]. In addition, such films can be loaded with drugs and act as systems for prolonged drug release at the sites of pathological processes [11,12,13]. In the food industry, antibacterial films are widely used to create active food packaging that significantly extends the shelf life of food products (fruits, vegetables, meat, fish, etc.) and has a significant economic effect [14,15,16]. In addition, some polysaccharide coatings can be used in edible food packaging [17].

Chitosan occupies a special place among a wide range of polymers that are used to obtain the mentioned films [18,19,20] due to its unique ecological, biological, and chemical properties. The ecological properties of chitosan lie in its biodegradability. This natural polymer completely and rather quickly decomposes in the natural environment (soil, water bodies, etc.); therefore, chitosan is a nature-friendly polymer and minimizes environmental burden [21,22,23]. The biological properties of chitosan include biocompatibility, biodegradability, non-toxicity, as well as antibacterial and antioxidant activities [4,24,25]. From a chemical point of view, chitosan is a linear copolymer of *N*-acetylglucosamine and glucosamine (Figure 1) [26,27,28]. The presence of a free amino group very favorably distinguishes chitosan from other polysaccharides and opens up wide possibilities for its chemical modification (in order to impart new attractive properties) [29,30,31].

Recently, nanoparticles (silver, montmorillonite, carbon nanostructures, etc.) have been widely used to improve the antibacterial properties of chitosan films and expand their mechanical characteristics (strength and elasticity) [32,33,34]. We recently prepared non-toxic active antibacterial nanoparticles of a complex of chitosan and Fe(III) ions [35], abbreviated as Fe(III)-CS-NPs. In this work, we focused on the elaboration of new chitosan-based films reinforced by Fe(III)-CS-NPs. In the first stage of this work, we decided to obtain films with different contents of Fe(III)-CS-NPs nanoparticles (2%, 5%, 10%, and 15%). In the second stage, we strengthened the film with the best mechanical and antibacterial properties by adding a small amount of a new cationic derivative of chitin in the hope that this strengthening could result in an even greater improvement in the mechanical and antibacterial properties of the film (although the authors are aware that the mechanical characteristics of chitosan-based films can be changed with storage time).

The new chitin derivative, in turn, was synthesized using the thiol–ene addition click reaction, which is widely used in modern organic chemistry as a so-called powerful reaction for the fast and convenient synthesis of a wide range of compounds [36,37].

The description of this study, its results, and outcomes are discussed in the sections that follow below.

## 2. Materials and Methods

In this study, we used crab shell chitin with a viscosity-average molecular weight (MW) of 17.2 × 10^4^ and a degree of acetylation of 100% (Sigma Aldrich, St. Louis, MO, USA). Chitosan with a viscosity-average molecular weight of 40 kDa, number-average molecular weight of 13.4 kDa, weight-average molecular weight of 51.2 kDa, polydispersity index of 3.8, and degree of acetylation of 15% was purchased from Bioprogress (Losino-Petrovsky, Russia). Rhodamine B was obtained from Aldrich (St. Louis, MO, USA). Other chemicals and solvents were obtained from commercial sources and used as received.

The Synthesis of the water-soluble cationic derivative of chitin (**CD**): Chitin (1.0 g) was dispersed in 35% NaOH solution (15 mL), then 15 mL of an aqueous solution of EDC (1.5 equiv.), NHS (1.5 equiv.), DMAP (0.5 equiv.), and but-3-enoic acid (1.5 equiv.) was added and stirred at room for 3 h. The reaction was quenched by the addition of acetone (30 mL), and the precipitate was washed with 30 mL of acetone/H_2_O (1/1) three times. The washed precipitate was dissolved in water with ca. 1.2 equiv. of thiocholine chloride, 0.2 equiv. of AIBN, and the reaction mixture was stirred at ca. 90 °C under argon for 2 h. The resultant polymer was precipitated by the addition of acetone (100 mL). The precipitate was dissolved in water, dialyzed against distilled water, and freeze-dried [38].

For blank film **O** (Table 1), chitosan (0.75 g) was dissolved in 30 mL 1% solution of acetic acid, then 0.45 g of glycerol was added. The resultant solution was cast on Petri dishes and dried at room temperature for a week.

For the preparation of **A2**, **B5**, **C10**, **D15** (see Table 1), 0.015 g, 0.0375 g, 0.075 g, and 0.01125 g of Fe(III)-CS-NPs, respectively, and glycerol (0.45 g) were added to the chitosan (0.75 g) solution in 1% CH_3_COOH (30 mL). To obtain film **Q**, 0.600 g of chitosan was dissolved in 30 mL of a 1% acetic acid solution, and then 0.150 g of **CD**, 0.075 g of Fe(III)-CS-NPs, and 0.450 g of glycerol were added. All films were dried at room temperature for a week [39].

The ^1^H NMR spectra were recorded on a Bruker Advance II spectrometer (Germany) operating at a frequency of 400 MHz.

The film thickness was measured with an electronic micrometer. The tensile strength and elongation at break were measured using RAM.1.1.A_RA (samples 6.0 cm long and 2.0 cm wide).

IR spectra were recorded on a Shimadzu IRSpirit at 4700 to 350 cm^−1^ (10 mg of sample without any specified sample preparation).

Differential Thermal Analysis (DTA) and thermogravimetric analysis (TGA) were recorded on SDT Q600 using a heating rate of 5 °C/min in the temperature range from 40 °C to 600 °C.

X-ray diffraction analysis was carried out on a Dron-7 X-ray diffractometer. With a 2θ angle interval from 7° to 40° with a scanning step ∆2θ = 0.02° and exposure of 7 s per point. Cu Kα radiation (Ni filter) was used, which was subsequently decomposed into Kα1 and Kα2 components during the processing of the spectra.

Antibacterial activity was evaluated completely as previously described elsewhere [38,40,41].

## 3. Results and Discussion

### 3.1. Preparation of the Films

Typically, “pure” chitosan-based films are characterized by high rigidity and brittleness. To overcome this shortcoming, it is customary to add a plasticizer to the molding solution. Glycerol proved to be one of the best plasticizers for chitosan-based films due to its high plasticizing efficiency and low cost. In the preliminary experiments, we determined the optimal content of glycerol in relation to the dry mass of the polymer matrix for a blank film, which amounted to 50–60%. This mass fraction of glycerol with respect to the total content of polymers and nanoparticles was used to prepare the films. A higher content of glycerol (65% or more) results in a deterioration in the mechanical properties (almost a twofold decrease in tensile strength).

Films with different contents of Fe(III)-CS-NPs nanoparticles [2% (film **A2**), 5% (film **B5**), 10% (film **C10**), and 15% (film **D15**)] were obtained by casting the solution. To completely dissolve chitosan in acetic acid, the solution was stirred continuously for 24 h. 

Lyophilized Fe(III)-CS-NPs nanoparticles easily redisperse in water, forming a transparent yellowish nanosuspension, restoring their typical hydrodynamic diameter (ca. 285 nm) and zeta potential (+31.8 mV). We added a calculated amount of the nanosuspension to the chitosan solution. This resulted in the formation of a light-yellow casting solution. Such molding solutions were obtained for the casting films.

Looking ahead, we have to report that among **A2**, **B5**, **C10**, and **D15**, mechanical and antibacterial properties were demonstrated in the case of film **C10**. Therefore, to obtain film **Q**, we used a **C10** film molding solution to which we decided to add 150 mg of a new cationic chitin derivative in the hope of obtaining a new film with even more attractive properties.

Therefore, the next stage of work was focused on the synthesis of a cationic derivative of chitin here and below abbreviated as **CD**. The decision to synthesize a cationic derivative of chitin was made because the introduction of a cationic substituent almost always dramatically enhances the antibacterial effect. The synthesis of the cationic chitin derivative **CD** is a two-stage process.

The first stage is the so-called pre-click modification, which consists of introducing a terminal alkene functionality into the chitin macromolecule. It would seem that for this purpose, it is convenient to use acrylic acid. However, we found that treating of chitin with acrylic acid does not result in the desired reaction, but Michael addition leads to the generation of a carboxyl derivative of chitin (Figure 1, B). Therefore, we took a different approach. It consisted of the treatment of chitin with but-3-enoic acid (using the DMAP-catalyzed method of activated esters in the presence of DMAP (dimethylaminopyridine), EDC, and *N*-hydroxysuccinimide (Figure 1, A)). In the but-3-enoic acid molecule, the double bond is an extremely weak Michael acceptor (in contrast to the double bond of acrylic acid). Therefore, heating a solution of chitin with but-3-enoic acid in the presence of DMAP, EDC, and *N*-hydroxysuccinimide at 60 °C led to the formation of the desired alkene derivative of chitin (Figure 1, A) [38,42].

The resulting alkene derivative of chitin was characterized by ^1^H NMR spectroscopy. The spectrum of the resulting polymer with the assignment of signals is shown in Figure 2. The degree of substitution of the alkene derivative of chitin was 0.65.

In the next stage, the resulting chiton alkene derivative was used as an ene component in the click reaction of thiol–ene addition with thiocholine, a thio analog of the natural compound “choline” (Figure 2).

The resulting polymer turned out to be soluble in water at room temperature over the entire pH range from strongly acidic (pH = 1) to strongly basic (pH = 14) and was characterized by ^1^H NMR spectroscopy. The spectrum of the resulting polymer with signal assignment is shown in Figure 3. The degree of substitution of the resulting polymer was 0.65. The degree of substitution of the polymer is equal to the degree of substitution of the starting alkene derivative of chitin, which indicates the completeness of the click reaction of thiol–ene addition.

Thus, cationic chitin derivative was introduced into the film molding solution, which allowed us to obtain film **Q**.

The content characteristics of the elaborated films are presented in Table 1 and their images are shown in Figure 4.

### 3.2. X-ray Diffraction Studies

X-ray analysis is an important method for studying the structure of film-like materials. The X-ray diffraction patterns of the resulting films are shown in Appendix A.

The diffraction patterns of the blank **O** film and the chitosan-based films reinforced by Fe(III)-CS-NPs **A2**, **B5**, **C10**, and **D15** are almost identical. Based on the results of X-ray diffraction studies, we attempted to assess the degree of perfection of the structure of the obtained films. The X-ray diffraction profiles of the amorphous peak were approximated using the pseudo-Voigt function. Thus, we refined the position, intensity, half-width, and the integral width of the peak.

As is known, integral broadening is related to the degree of perfection of the structure or broadening due to the size of micro- or nanoblocks. The smaller the integral broadening, the more perfect the structure [43]. Table 2 demonstrates the obtained characteristics of the integral broadening based on the results of refinement of the profile of the amorphous peaks of the studied samples. Blank film **O** and films **A2** and **B5** had a greater integral broadening (worse structure) than films **C10** and **D15**. In general, we observed a decrease in the integral broadening of the amorphous peak as the concentration of Fe(III)-CS-NPs increased, which also correlated with the obtained mechanical properties (see the next section). However, the mechanical characteristics of the film with **D15** turned out to be worse compared to the **C10** film.

The diffraction pattern of the chitosan **Q** film (which contains **CD**) differs significantly from all other diffraction patterns (see Appendix A). In addition to the amorphous peak, the diffraction pattern of film **Q** exhibits peaks in the 2θ region: 8.24°, 11.24°, 15.8°, and 17.96°, which can be explained by the presence of the cationic chitin derivative in the polymer matrix. Film **Q** is characterized by the greatest perfection of the structure. The crystallographic parameters of the **Q** film are presented in Table 3.

### 3.3. FTIR Spectroscopy

IR spectroscopy is based on the absorption of infrared light by a substance, and it is commonly used to estimate the chemical structure of films.

The IR spectra of all analyzed films were practically identical and display stretching vibrations bands characteristic for chitosan, i.e., wide bands of O–H and N–H stretching (3440–3100 cm^−1^), C–H stretching (2870 cm^−1^) and bending (1460, 1420, and 1380 cm^−1^) vibrations, and N–H deformation vibrations (1590–1650 cm^−1^). Absorption bands in the range of 900–1200 cm^−1^ are due to C–O–C, C–C, and N–H deformation vibrations. The spectra also showed bands characteristic for protonated NH_3_^+^ group and CH_3_COO^−^ (1300–1640 cm^−1^) [44]. The spectrum of **Q** had the characteristic absorption of ^+^N(CH_3_)_3_ at 1477 cm^−1^ [45] (Figure 5).

The spectra of **A2**, **B5**, **C10**, and **D15** do not show chitosan-Fe(III) characteristic bands [35] due to their small concentrations (Figure 5).

### 3.4. Mechanical Properties of the Films

The most important mechanical properties of film-like materials are tensile strength and elongation at break [46]. These mechanical parameters are usually strongly dependent on the concentration and chemical structural features of the constituents of the tested film (polymers that make up the polymer matrix, plasticizers, and fillers). The tensile strength is regarded as the maximum stress that a material can withstand while being stretched or pulled before breaking. The elongation at break is the ratio of the initial and final lengths of the film before it breaks. Thus, tensile strength is a measure of film strength, while elongation at break characterizes the elasticity of the film material. Pure chitosan films are characterized by high strength but very low elasticity; therefore, they are dramatically brittle. To avoid this drawback, plasticizers, such as glycerin, are used.

Figure 6 and Figure 7 demonstrate the results of the mechanical tests of the elaborated films.

The blank film **O** demonstrates the lowest tensile strength and elongation at break. Adding Fe(III)-CS-NPs leads to an increase in both strength and elasticity. Films **A2**, **B10**, and **D15** have almost identical values of tensile strength and elongation at break. Film **C10** demonstrates a pronounced increase in both the strength and elasticity compared to blank film and films with other Fe(III)-CS-NPs containing films. Thus, adding quaternized chitosan derivative **CD** to film **C10** (which results in film **Q** formation) leads in 30% to an increase in tensile strength and to an almost two-fold increase in elasticity of **Q** compared to **C10**. This may be because the sterically hindered substituents of polymer **CD** introduced into the chitosan polysaccharide backbone destroy the systems of interchain hydrogen bonds and act as plasticizers. Moreover, sample **Q** has a slightly higher content of glycerol, which may have some impact on the mechanical properties.

A comparison of the elaborated polymer films with those reported in the literature shows that, in fact, our films exhibit lower tensile strength. When evaluating the elongation at break parameter, the elaborated films demonstrated similar characteristics to those described in the literature. However, it is worth noting that while film **Q** has moderate tensile strength, it exhibits an exceptionally high elongation parameter [47,48,49].

### 3.5. TGA/DTA Analysis

The blank film **O** TGA curve has three stages of thermal degradation (Figure 8). The first stage is associated with a small weight loss due to the evaporation of the absorbed and weakly bound water (mass loss 8%, T_max_ = 150 °C). The second stage and the third stage are associated with the degradation of the polymer structure (mass loss 44%, T_max_ = 380 °C for the second stage and mass loss 47% and T_max_ = 520 °C for the third stage). The second stage was accompanied by a moderate exothermic effect, and the third stage had a spasmodic pronounced exothermic effect. 

Figure 8 also displays the thermal degradation patterns of **A2**. The stage of water loss corresponds to a weak endothermic effect (mass loss 10%, T_max_ = 105 °C). The second exothermic effect was observed at T_max_ = 380 °C and is associated with a mass loss of 56%. The third stage of decomposition was accompanied by a strong exothermic effect (mass loss 34%, T_max_ = 530 °C) (Figure 8 and Figure 9). 

The TGA curves of **B5**, **C10**, **D15**, and **Q** are almost identical, and the values of weight loss and maximum temperature differ by no more than 10%. The first stage of thermal degradation was associated with water loss (average mass loss 9%, average T_max_ = 105 °C). The second stage was accompanied by an acute exothermic effect (average mass loss 64%, average T_max_ = 400 °C), followed by a gradual loss of mass occurs. The third stage was accompanied by a sharp pronounced exothermic effect (average T_max_ = 540 °C, average weight loss 27%), and **C10** curve demonstrates double exothermic peak at this stage (Figure 8 and Figure 9).

### 3.6. Antibacterial Activity

Chitosan-based films are usually of interest for biomedical applications for treating wounds or burns, as well as in the food industry for the so-called active packaging of products in order to extend their shelf life. In both cases, these applications of chitosan-based films are possible due to their antibacterial properties. In this work, we studied the antibacterial activity of the elaborated films.

The results of the in vitro antibacterial tests (Table 4) clearly show that the antibacterial activity of the Fe(III)-CS-NPs reinforced films **A2**, **B5**, **C10,** and **D15** is higher than the antibacterial effect of the blank film **O**. Moreover, the antibacterial activity of the mentioned films is strongly dependent on the concentration of Fe(III)-CS-NPs in the films. However, this relationship was not strictly linear. The greatest increase in the antibacterial effect with an increase in the content of Fe(III)-CS-NPs was observed up to an increase in the concentration of Fe(III)-CS-NPs from 0 to 10% (**O**, **A2**, **B5**, **C10**). When the concentration increased from 10% (**C10**) to 15% (**D15**), only a moderate increase in the antibacterial effect was observed, although the concentration of Fe(III)-CS-NPs also increased significantly. Thus, for the antibacterial effect, the concentration of Fe(III)-CS-NPs was 10% (film **C10**), which was established as the optimal concerntration. 

It is for this reason that we decided to introduce the polymer **CD**, which has a high cationic density, into the **C10** film in order to further increase the antibacterial activity. Thus, we obtained film **Q**. Indeed, film **Q** is characterized by the highest antibacterial activity among all elaborated films (inhibition zone 19.6 ± 0.2 for *S. aureus* and 14.2 ± 0.1 for *E. coli*). The high antibacterial activity of film **Q**, we believe, is due to several factors, including the presence of (i) chitosan in the film, which itself possesses an antibacterial effect, (ii) antibacterial Fe(III)-CS-NPs nanoparticles with a high positive zeta-potential and, finally, (iii) cationic derivative of chitin **CD**. The symbatic action of all the mentioned components of film **Q** is certainly able to result in a pronounced increase in the antibacterial activity of **Q** in comparison with other elaborated films. It is also worth noting that the glycerol included in the films possesses moderate antibacterial properties [50].

To date, there is no consensus on the mechanism of the antibacterial effect of chitosan and its based systems, which is associated with its multiple non-specific actions. For Gram-positive bacteria, this can be the interaction of chitosan with negatively charged murein components, teichoic and teichuronic acids, which, in turn, can induce hyperactivation of autolytic enzymes, which are normally deposited on these polyanions and subsequently cause uncontrolled degradation of the cell wall glycoprotein. In Gram-negative bacteria, this is the interaction of chitosan with negatively charged phosphate groups of polysaccharides located on the outer membrane of the bacterial cell [51,52,53].

In general, the electrostatic interaction of the chitosan polycation with the negatively charged areas of the microbial cell surface leads to at least two events that are unfavorable for the microorganism:

(1) a change in membrane permeability that provokes a pronounced osmotic imbalance;

(2) hydrolysis of peptidoglycans of the microbial cell wall, leading to the loss of many intracellular compounds (proteins, nucleic acids, and lactate dehydrogenase) [51,54].

Electron microscopy studies show that at the site of damage, the cell membrane thickens, in some places it has a very blurred appearance, near such areas one can observe accumulations of agglutinated substances, probably representing components of the outer membrane or outward periplasmic or cytoplasmic contents. The probability of the release of cytoplasmic content is confirmed by the fact that cell protoplasts in the presence of chitosan are greatly reduced in size, and agglutinated components on the cell surface are concentrated in the area where the remains of cytoplasmic content are adjacent from the inside to the outer membrane [54].

Our previous studies have shown that a very convenient way to monitor the integrity of the membrane is spectrophotometry of a suspension of bacterial cells in a 0.5% aqueous solution of sodium chloride in the UV region [35]. This approach is based on the fact that intracellular components are characterized by strong absorption at 260 nm. Through this method, in this work, we studied the effect of the elaborated films on the integrity of the cell membranes of *S. aureus* and *E. coli*.

Both *E. coli* and *S. aureus* suspensions (Figure 10 and Figure 11) in the presence of the elaborated films showed an increase in optical density at 260 nm for about 50 and 20 min of the experiment, respectively, after which a plateau was observed. When using Fe(III)-CS-NPs reinforced chitosan-based films (**A2**, **B5**, **C10**, **D15**, and **Q**), regardless of the concentration of Fe(III)-CS-NPs, the time to reach a plateau is approximately the same as that for blank film **O**. However, the value of the optical density of A2_60_ is higher for all Fe(III)-CS-NPs reinforced chitosan-based films compared to blank film **O** and has a tendency to increase with an increasing concentration of Fe(III)-CS-NPs. We found that the most effective film was **Q**. This fact is not surprising, since this film contains the cationic derivative of chitin **CD** [55]. The increased cationic density of the polymer, in turn, always inevitably leads to an increase in its antibacterial activity [56]. The obtained experimental data indicate that the elaborated films are effective in causing membrane integrity disruption in both *E. coli* and *S. aureus*, and this damage to the bacterial membranes is a good explanation for the antibacterial effect of the films.

## 4. Conclusions

The results of this work can be considered from the following perspectives.

Firstly, we elaborated new Fe(III)-CS-NPs reinforced chitosan-based films with different concentrations of the mentioned nanoparticles. Nanoparticles Fe(III)-CS-NPs improved both the mechanical and antibacterial characteristics of chitosan-based films. We confirmed that the optimum concentration of Fe(III)-CS-NPs for the improvement of antibacterial and mechanical properties is 10%.

Secondly, in the framework of this study, we have obtained a new water-soluble cationic derivative of chitin **CD**, using the thiol–ene click reaction in chitin chemistry. The addition of **CD** to the chitosan matrix of the elaborated film significantly improves its mechanical and antimicrobial properties.

Thirdly, we confirmed that the key mechanism for the antibacterial action of the obtained films is disruption of the membranes of bacterial cells.

Finally, the elaborated antibacterial films are of interest for potential biomedical and food applications, and this project is underway b our group.

Of course, this study has some limitations related to the development of industrial as well as calculation improvement of economic feasibility, which is intriguing for further research.

## Data Availability

Not applicable.

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
