# Peer review of "Novel Highly Efficient Antibacterial Chitosan-Based Films"

_biotech, 2023, doi:10.3390/biotech12030050_

Round 1

Reviewer 1 Report

Overall, the work is planned and executed well, but presentation of results and manuscript preparation should be improved. Please go through the comments and revise the manuscript accordingly.

Should be improved, in certain cases: sentence should be rewritten and grammatical mistakes should be rectified.

Author Response

The authors are sincerely grateful to the Reviewer for his unselfish and extremely important work, for thoroughly checking the manuscript, for the valuable comments, which have significantly improved the starting text of the submitted manuscript.

Abstract

  1. Line 13: “The addition of this chitin derivative to the polymer matrix to the film significantly improves its mechanical and antimicrobial properties”.

- Corrected.

  1. Line 15: Rewrite the sentence.

- Corrected.

  1. Line 16-17: Rewrite the sentence.

- Corrected.

  1. Provide the data regarding the experimental results.

- Corrected.

  1. Abstract is too generic and needs significant restructuring of the information and details of the experiment results.

- Corrected.

  1. Key Contribution: It just repeats what was already stated in the abstract. Abstract should focus on the aim and results obtained, so that key contribution can focus on the novelty of the study and outcome.

- Corrected.

Introduction

  1. Line 25: ‘relentless’ is not the right word.
    - Corrected.
  2. Line 32: Please provide few commercially available “cling films”.
    - Corrected.

Materials and Methods

  1. Line 77: What is the meaning of “O preparation”?

- Corrected.

  1. Please provide details regarding “O, A2, B5, C10, D15, and Q” in section 2.

- Corrected.

  1. Line 77, 82: “mL”.

- Corrected.

  1. Line 99-100: Please rewrite the sentence. Remove “by some of us”.

- Corrected.

  1. Please cite proper references for the synthesis of chitin derivative and film preparation.

- This chitin derivative was obtained by us for the 1st time. We have extensive experience in synthesis of chitin derivatives (10.1016/j.ijbiomac.2022.04.199; 10.1016/j.foodchem.2020.128696; 10.1016/j.ijbiomac.2020.09.123; 10.1016/j.mencom.2022.05.022; 10.1016/j.carbpol.2020.117167; 10.1016/j.carbpol.2020.117593). We provided reference to our previously published works (thiol-ene modification of chitosan) and reference to preparation of chitosan/glycerol films.

  1. Rewrite the Methodology section, its entirely vague and does not give complete details regarding the preparation.

- The methodology section is written in sufficient volume to reproduce the experiment (for specialist). We will be very grateful to the reviewer if he allows to leave it as it is.

Results and Discussion

  1. Line 105: “To completely dissolve chitosan in acetic acid, the solution was stirred continuously for 24 h”.

- Corrected.

  1. Line 103-104: A2 is for 2% NPs, B5 for 5% NPs, and so on. But in Line 80, A2 is weighed 0.15 g and B5 about 0.0375 g. As the percentage of NPs increased, so does the amount weighed or added to the solution. But in this case, the amount weighed decreased, as the percentage of NPs to be added to the solution increased.

- Thank you! We lost one «0». Of course, 0.015 g.

  1. Line 121-132: Please cite proper references.

- Corrected.

  1. Include more references while discussing the results obtained.

- Corrected.

  1. Should provide the Supplementary Data for reviewing.

- We uploaded Supplementary Materials.

  1. Also include the XRD data in the manuscript.

- They are in Supplementary Materials.

  1. Include Antibacterial plate images in the manuscript.

- Unfortunately, our colleagues who are engaged in biological experiments do not make photographic documentation. This experiment has already been completed.

  1. Should perform Electron Microscopy and EDX analysis for better characterization of films.
  • EDX spectra are added in Supplementary Materials.

Conclusion

  1. Conclusion should summarize the study focusing on the outcomes achieved rather than repeating the same abstract and key contributions.

- Corrected.

References

  1. Should include more relevant literature in Methodology and Discussion sections.

- Corrected.

Reviewer 2 Report

More details with characteristics of chitosan would be beneficial.

The terms "highly antibacterial" in lines 16 and 346, in the titles of published this 2023 year in journals Processes and Catalysts cited articles of the same authors, "highly active" in line 50, "highly soluble" in line145, have not been presented quantitatively and are not determined specifically. The terms are not generally accepted, they are too general and are not specific, so there is need too characterize and identify them in much more detail. The difference between antibacterial and highly antibacterial  must be clearly and quantitatively explained.

The manuscript lacks the comparison of mechanical properties of chitosan films with published data from literature. The strength of chitosan films is lower compared with some presented in published literature data. 

The change in mechanical properties, including the improvement of mechanical properties with storage time, has not been  taken into account and not discussed in INTRODUCTION, as well as not taken into account and not discussed published articles related with the strong effect of glycerol concentration on mechanical properties. The used in manuscript selected content of glycerol in films has not been duly justified. The Figures 6 and 7 show essential increase in mechanical properties of sample Q which has a bit higher content of glycerol content compared with other samples.

Why in lines 274 - 283 the antibacterial potential of glycerol  has not been also discussed?

The curves in Figures 8 and 9 are to a great extent overlapping and cannot be properly distinguished.

The more detailed discussion of part 3.3 of infrared spectra would be beneficial.

In CONCLUSIONS in line 344 it would be correctly instead of "we found" to write "we confirmed".

small corrections are necessary somewhere, e.g. in line 71

Author Response

The authors are sincerely grateful to the Reviewer for his unselfish and extremely important work, for thoroughly checking the manuscript, for the valuable comments, which have significantly improved the starting text of the submitted manuscript.

More details with characteristics of chitosan would be beneficial.

- Corrected.

The terms "highly antibacterial" in lines 16 and 346, in the titles of published this 2023 year in journals Processes and Catalysts cited articles of the same authors, "highly active" in line 50, "highly soluble" in line145, have not been presented quantitatively and are not determined specifically. The terms are not generally accepted, they are too general and are not specific, so there is need too characterize and identify them in much more detail. The difference between antibacterial and highly antibacterial  must be clearly and quantitatively explained.

- Corrected.

The manuscript lacks the comparison of mechanical properties of chitosan films with published data from literature. The strength of chitosan films is lower compared with some presented in published literature data. 

- Corrected.

The change in mechanical properties, including the improvement of mechanical properties with storage time, has not been  taken into account and not discussed in INTRODUCTION, as well as not taken into account and not discussed published articles related with the strong effect of glycerol concentration on mechanical properties. The used in manuscript selected content of glycerol in films has not been duly justified. The Figures 6 and 7 show essential increase in mechanical properties of sample Q which has a bit higher content of glycerol content compared with other samples.

- Corrected.

Why in lines 274 - 283 the antibacterial potential of glycerol  has not been also discussed?

- Corrected.

The curves in Figures 8 and 9 are to a great extent overlapping and cannot be properly distinguished.

- Yes, but we can`t to change them.

The more detailed discussion of part 3.3 of infrared spectra would be beneficial.

- The spectra are very similar, we will be cordially grateful to Reviewer if he allows it as it is.

In CONCLUSIONS in line 344 it would be correctly instead of "we found" to write "we confirmed".

- Corrected.

Reviewer 3 Report

The authors elaborated new chitosan-based films reinforced by iron-coating chitosan nanoparticles Fe-CS-NPs with different concentration and evaluated the mechanical and antimicrobial properties of the obtained films. The manuscript falls within the scope of the journal and the authors have been through in presenting their experimental data. However, some major concerns should be addressed before considering publishing.

In the “Introductin” part, it would be helpful if the authors could provide a bit more about the expected benefits or advantages of incorporating Fe-CS-NPs and the chitin derivative into the films which could further emphasize the potential significance of the study. 

Several articles published within the last five years serve as valuable references and would enhance the depth and comprehensiveness of your work: doi.org/10.1016/j.micromeso.2020.110113, doi.org/10.1016/j.mtcomm.2022.104874, doi: 10.3389/fbioe.2022.989729, doi.org/10.2147/IJN.S212807) Considering these references would enhance the depth and comprehensiveness of your work.

Please add the discussion of the significance of this study and how could these films address current challenges and contribute to advancements in this field?

In order to provide a more comprehensive and impactful summary of the study’s conclusions, authors should add the limitations of this study instead of list all the results in the “Conclusion” section.

N/A

Author Response

The authors are sincerely grateful to the Reviewer for his unselfish and extremely important work, for thoroughly checking the manuscript, for the valuable comments, which have significantly improved the starting text of the submitted manuscript.

In the “Introductin” part, it would be helpful if the authors could provide a bit more about the expected benefits or advantages of incorporating Fe-CS-NPs and the chitin derivative into the films which could further emphasize the potential significance of the study. 

- Corrected.

Several articles published within the last five years serve as valuable references and would enhance the depth and comprehensiveness of your work: doi.org/10.1016/j.micromeso.2020.110113, doi.org/10.1016/j.mtcomm.2022.104874, doi: 10.3389/fbioe.2022.989729, doi.org/10.2147/IJN.S212807) Considering these references would enhance the depth and comprehensiveness of your work.

- Corrected.

Please add the discussion of the significance of this study and how could these films address current challenges and contribute to advancements in this field?

- Corrected.

In order to provide a more comprehensive and impactful summary of the study’s conclusions, authors should add the limitations of this study instead of list all the results in the “Conclusion” section.

- Corrected.

Round 2

Reviewer 2 Report

The more detailed characterization of chitosan is necessary.

The results for films with  higher content of glycerol  are necessary

Acceptable

Author Response

The more detailed characterization of chitosan is necessary.

- Corrected.

The results for films with  higher content of glycerol  are necessary

- Corrected.

Reviewer 3 Report

Congratulations to your excellent work!

Author Response

Thank you!